# How Do Quorum-Sensing Signals Mediate Algae–Bacteria Interactions?

**DOI:** 10.3390/microorganisms9071391

**Published:** 2021-06-27

**Authors:** Lachlan Dow

**Affiliations:** Root Microbe Interactions Laboratory, Australian National University, Canberra 0200, Australia; lachlan.dow@anu.edu.au

**Keywords:** microalgae, marine bacteria, quorum-sensing signals, alkyl quinolones, microbial loop

## Abstract

Quorum sensing (QS) describes a process by which bacteria can sense the local cell density of their own species, thus enabling them to coordinate gene expression and physiological processes on a community-wide scale. Small molecules called autoinducers or QS signals, which act as intraspecies signals, mediate quorum sensing. As our knowledge of QS has progressed, so too has our understanding of the structural diversity of QS signals, along with the diversity of bacteria conducting QS and the range of ecosystems in which QS takes place. It is now also clear that QS signals are more than just intraspecies signals. QS signals mediate interactions between species of prokaryotes, and between prokaryotes and eukaryotes. In recent years, our understanding of QS signals as mediators of algae–bacteria interactions has advanced such that we are beginning to develop a mechanistic understanding of their effects. This review will summarize the recent efforts to understand how different classes of QS signals contribute to the interactions between planktonic microalgae and bacteria in our oceans, primarily *N*-acyl-homoserine lactones, their degradation products of tetramic acids, and 2-alkyl-4-quinolones. In particular, this review will discuss the ways in which QS signals alter microalgae growth and metabolism, namely as direct effectors of photosynthesis, regulators of the cell cycle, and as modulators of other algicidal mechanisms. Furthermore, the contribution of QS signals to nutrient acquisition is discussed, and finally, how microalgae can modulate these small molecules to dampen their effects.

## 1. Introduction

### 1.1. The Phycosphere

Marine microalgae are a highly diverse array of photosynthetic organisms, comprised of prokaryotic (i.e., cyanobacteria) and eukaryotic taxa. Eukaryotic microalgae stem from each of the putative endosymbiotic plastid lineages [1], such as primary (green algae), secondary (red algae), and tertiary endosymbionts (diatoms, haptophytes and dinoflagellates). Microalgae live planktonic lifestyles (suspended as single-celled organisms or in small colonies) or benthic lifestyles (colonizing surfaces). Planktonic microalgae are the foremost primary producers in the ocean, and are principal components of the earth’s biogeochemistry [2,3,4]. However, they live among marine bacteria whose effects on microalgae are both diverse and pervasive. It is estimated that bacteria comprise up to 25% of the oceans’ biomass [5], vastly outnumbering algae on a per cell basis. Bacteria in aquatic systems respire between 25 and 50% of the carbon fixed by algae, and this carbon is often metabolized by bacteria mere hours after it has been fixed [5,6]. Some of these bacteria are ‘passive’ players, surviving off nutrients exuded by microalgae [7], while other bacteria are ‘algicidal’ and have evolved methods to inhibit the growth or even kill microalgae, thereby leading to cell lysis and a spike in the nutrients in their immediate vicinity [8]. This cyclical process of fixation of carbon dioxide by algae and respiration by bacteria forms the basis of the microbial loop [9], an important step in the global carbon pump. However, as research has uncovered the ‘chemical currencies’ interchanged between bacteria and microalgae [10], the microbial loop has resolved into a microbial network, in which chemical cues influence the biology of algae and bacteria. This review discusses certain chemicals, called quorum-sensing signals, which are produced by specific marine bacteria. Specifically, it addresses ways in which bacteria utilize these signals during interactions with planktonic microalgae, both directly as algicidal chemicals, and indirectly, for example, to regulate other algicidal mechanisms. 

Microalgae from natural samples are typically associated with bacteria, which are either attached to or closely associated with the alga. These populations of algae-associated bacteria differ from those found freely suspended in the water column [11,12]. Algae cells are also surrounded by a ‘boundary layer’: a film of water that does not mix readily with the surrounding water, and thus has a distinct physical makeup compared to the bulk properties of open water (reviewed by Seymour et al. [13]). Both this unique cohort of bacteria and physical properties of an alga cell forms the basis of the phycosphere: a unique microenvironment surrounding algae, which is distinct from the rest of the water column. It is within the phycosphere that the majority of interactions between algae and bacteria take place [14].

### 1.2. Quorum-Sensing Compounds and Their Relatives

Quorum sensing (QS) refers to a type of ‘molecular circuit’, which some species of bacteria use to conduct intercellular signaling. Genetically, QS requires at least two sets of genes: one or more synthase genes, which synthesize a small diffusible molecule known as a QS signal or autoinducer, and one or more receptors, which detect the presence of the QS signal. The activity of this receptor protein dictates the transcription of a wide variety of genes based on the concentration of QS signal it detects, including synthesis of the signal itself, hence the term autoinducer. Thus, gene expression governed by quorum sensing is dependent on the local concentration of QS signals surrounding the bacterium. In this way, bacteria have the ability to coordinate gene expression and physiological processes in response to cell density [15]. These processes typically include biofilm formation and motility [16], and in the case of microalgae-associated bacteria, QS signals can regulate interactions between bacterium and alga. 

QS signals can be categorized by their chemical structure. While there are a great variety of QS systems and corresponding autoinducers, this review will primarily focus on the *N*-acyl-homoserine lactone (AHL, Figure 1 1–3) and 2-alkyl-4-quinolone (AQ, 5–8) compound families, as these regularly play roles in algae–bacteria interactions. I will also briefly discuss the QS signal AI-2, a furanosyl borate diester (9 Figure 1). The AHL and AQ quorum-sensing signals display a range of structural variation, depending on the species. The AHL is a lactone moiety bound to an *N*-linked acyl side chain. This acyl chain varies in length from four to over eighteen carbon atoms. The acyl chain can possess double bonds, and the third acyl chain carbon can possess a hydroxyl or keto group. In this article, AHL compounds are named to reflect this variation, for example, 3-oxo-C12-AHL corresponds to an AHL with a twelve-carbon acyl chain and a keto group modification, as shown in Figure 1. This keto modification is a particularly important feature in algae–bacteria interactions, because it facilitates a rearrangement reaction leading to tetramic acids (TA, 4) [17]. AQ molecules from bacteria also vary regarding the length and saturation of their alkyl side chain, as well as the oxidation state of the nitrogen atom, and are named as such, for example, 2-nonyl-quinolone *N*-oxide (7), shown in Figure 1. The “Pseudomonas quorum signal” (6, PQS) is a unique AQ signal possessing a hydroxyl group at the third carbon, and a heptyl alkyl side chain (Figure 1). It should be noted that there is considerable structural diversity of AQs in nature, which are beyond the scope of this review [18,19]. 

Bacteria isolated from marine sources often possess QS systems [20]. Although the first discovery of QS was from a marine source [21], it was some time until QS in marine bacteria was considered more routinely [22]. It is now clear that a diverse range of marine bacteria produce QS signals. AHL and AQ signals are regularly isolated from marine bacteria, especially taxa belonging to the Roseobacter clade, the *Pseudomonas*, *Alteromonas*, *Pseudoalteromonas* and *Vibrio* genera [23,24,25,26,27,28,29]. Furthermore, because QS can dictate virulence signals, particular attention has been paid to QS in *Vibrio* spp. as species of this genus often have multiple QS circuits [26], including *Vibrio* species which are pathogens of human [30], fish, and mollusks [31,32,33]. 

It is now clear that along with being an intra-species signal, QS signals often act as inter-species and inter-kingdom effectors, in a wide range of ecosystems [34,35,36,37,38]. In the case of planktonic microalgae and bacteria, the phycosphere is a microcosm in which quorum-sensing signals play an important role. Research in marine systems has progressed such that we are now gaining a mechanistic insight into how QS signals mediate algae–bacteria interactions. These interactions may play out on a microscopic level, but also represent one of many connections within the large scale of the microbial loop (many of these connections are discussed by Cirri and Pohnert [39]). In this review, I shall discuss how certain QS signals directly affect the growth of planktonic microalgae, as well as how quorum sensing can dictate other microbial processes thus indirectly dictating algae–bacteria interactions, and the microbial loop as a whole.

## 2. Direct Effects of Quorum-Sensing Signals on Microalgal Growth and Metabolism

### 2.1. N-Acyl Homoserine Lactones

Much research has been devoted to identifying AHLs from marine bacteria, and their effects on microalgae are varied and often subtle. AHLs are frequently isolated from microalgae-associated bacteria, and these bacteria often produce a diverse array of AHLs [25,40,41]. For example, Ziesche et al. conducted a chemical survey of marine Roseobacter isolates, identifying a wide variety of AHLs. Although 19 different AHLs were detected in bacterial extracts, only one AHL, (7Z)-C14:1-AHL produced by *Roseovarius mucosus* isolated from a dinoflagellate culture, inhibited diatom growth [40]. In another study, contrasting effects of AHLs on the diatom *Seminavis robusta* were reported [42]. Namely, Stock et al. observed a growth promotion with C14-AHL treatment, while treatments with OH-C14-AHL and oxo-C14-AHL inhibited growth. The authors also provided transcriptomic evidence describing how diatoms respond to AHLs. Among other responses, diatoms altered glutathione metabolism transcription to both AHL treatments tested (C14-AHL and oxo-C14-AHL). Furthermore, genes involved in lipid metabolism were differentially expressed in response to oxo-C14-AHL, as were cell cycle genes, hinting at the mechanisms at play behind diatom growth inhibition. Although *S. robusta* is a benthic diatom, the structural homolog oxo-C12-AHL also suppressed growth in the planktonic *Phaeodactylum tricornutum* [43]. These results shed light on how microalgae respond to AHLs in a diverse and structure-specific manner.

### 2.2. Tetramic Acids

While the effects of AHLs on microalgae are relatively subtle, certain AHLs spontaneously undergo an enzyme-free Claisen-like rearrangement to produce tetramic acids. This reaction requires a 3-keto moiety, and has been observed in artificial seawater [17,43]. This ‘secret weapon’ of oxo-AHLs has direct consequences for microalgal growth. An impressive diversity of structurally-related tetramic acids have also been isolated from a variety of marine microbes [44]. 

In 2019, Stock et al. reported the algicidal effects of the tetramic acid ‘TA12’ (**4**, Figure 1), formed by rearrangement of oxo-C12-AHL [43]. Flow cytometry revealed extensive diatom mortality after several days’ exposure to TA12 in the diatom *P. tricornutum*. Pulse-amplitude-modulation (PAM) fluorometry showed that photosynthetic electron transport was inhibited within ten minutes of TA12 addition. Fast fluorescence transients pinpointed the source of this inhibition to Photosystem II (PSII) turnover. Further analyses identified even more potent effects with TA14 (the rearrangement product of oxo-C14-AHL). In a subsequent study, 5 µM TA14 suppressed the growth and photosynthetic efficiency of the diatom *Seminavis robusta* [42]. Transcripts involving lipid metabolism were differentially expressed, similar to treatment with the precursor AHL (see above), however, TA14 treatment did not induce a change in cell-cycle gene transcription, thus distinguishing the effects of these two molecules on diatom physiology. These studies have uncovered an additional layer of how oxo-AHLs affect microalgae, via the rearrangement of the AHL to TAs. 

### 2.3. 2-alkyl-4-quinolones

The effects of AQs on photosynthesis were discovered before they were identified as QS signals [45,46]. So too were they isolated from marine bacteria before their role in QS was discovered [47]. However, it is only recently that these aspects of AQ biology have been considered simultaneously. In comparison to the various effects of AHLs on microalgae, AQs are most often algicidal. Intriguingly, only a small subset of AQs are QS signals, while many AQs act as growth inhibitors during interspecies interactions.

AQs have been repeatedly isolated from marine bacteria, and have been shown to inhibit the growth of microalgae. For example, the AQ pentyl-quinolone has been isolated from species of *Alteromonas* and *Pseudoalteromonas,* and have been shown to inhibit diatom growth [48,49,50]. In 2012, Cho isolated *Alteromonas* sp. KNS-16 from a large blooming event of the harmful alga *Heterosigma akashiwo* (a Raphidophyte) [24]. After observing the algicidal effects of this bacterium, they isolated two unusual AQs, 1′E-undecenyl-quinolone (8, Figure 1) and undecyl-quinolone, and displayed their strong algicidal effects. The half-inhibitory concentration (LC_50_) of 8 was less than 5 µM against *H. akashiwo* and two dinoflagellates, while when tested on the green alga *Tetraselmis suecica*, 8 was considerably less toxic. Green algae are apparently less sensitive to AQs, as was shown by Patidar et al., who found that the inhibitory effects of AQs produced by *Pelagibaca bermudensis* were outweighed by other growth promoting effects when co-cultured with the green alga *Tetraselmis striata* [51]. Other AQs have been isolated from a variety of marine environments, including several unusual AQs from sponge-associated bacteria [52,53,54] and heptyl-quinolone from planktonic and algae-associated bacteria [27,55,56]. 

The AQ heptyl-quinolone (HHQ) is a particularly potent algicide with several modes of action. In 2016, Harvey and coworkers isolated HHQ from the marine species *Pseudoalteromonas piscicida*, displaying striking toxicity (IC_50_ < 1 µM) for the coccolithophore *Emiliania huxleyi* [27]. HHQ has since been intensively studied, and we are beginning to attain a detailed understanding of its effects on algae, and the phycosphere in general. A thorough physiological examination has shown that HHQ inhibits diatom growth by obstructing photosynthetic electron transport, via blockage of the cytochrome *b*_6_*f* complex [57]. However, in the green alga *Dunaliella tertiolecta*, photosynthesis was inhibited via inactivation of PSII, and in *E. huxleyi*, inhibition was achieved via both PSII and cytochrome *b*_6_*f* inhibition. Further observations also showed that HHQ simultaneously impacts respiration, identifying a two-pronged mechanism of action against diatom growth. In another study, Pollara et al. investigated the molecular effects of nanomolar concentrations of HHQ on *E. huxleyi* [58]. Among other effects, they discovered HHQ arrests the cell cycle and DNA maintenance, indicating that HHQ inhibits microalgae growth via a third mechanism. The inhibition of cell cycle, including the downregulation of cell cycle genes is reminiscent of the results observed by Stock et al. upon treatment of diatoms with oxo-C14-AHL [42]. Pollara et al. also observed an increased resistance against viral infection in *E. huxleyi* treated with HHQ, hinting that HHQ may provide further services when considering more actors within the phycosphere. It is so far unclear if these modes of action occur concurrently with photosynthesis and respiration inhibition, although electron microscopy experiments conducted by Pollara et al. did observe changes to plastid architecture during HHQ treatments, as well as differential expression of genes involved in photosynthesis [58]. However, it is clear that AQs, along with AHLs and tetramic acids, play a role independent of quorum sensing, and can have direct consequences on microalgae growth and proliferation. While the effects of AQs and TA on photosynthesis are due to direct interactions between the respective QS molecule (which has permeated the alga cell) and thylakoid protein, the responses observed by both Pollara and Stock hint at some regulatory ability of microalgae in response to QS signals [42,59]. 

Many QS signal-producing bacteria (particularly those producing AQs) have been isolated from algal blooms [24,60], an environment in which inhibiting algal growth would be a sensible strategy from the bacterium’s point of view. When algae cell densities are high, such as during algal blooms, a few more dead algae does not jeopardize the bacterium’s carbon source (photosynthesizing microalgae), while the lysed cells allow bacteria to capitalize on the liberated nutrients (such as N, P and Fe), especially as the bloom wanes and nutrients begin to become scarce. In turn, algicidal compounds may ensure bacteria are not overcome by antimicrobials produced by the microalgae. Furthermore, QS allows bacteria to respond to bursts in nutrient supply, as shall be discussed below.

## 3. Indirect Effects of Quorum-Sensing Signals on Microalgae: Algae–Bacteria Interactions Regulated by Quorum Sensing 

So far, this review has focused on how QS signals act as a direct signal between bacteria and microalgae. However, it is clear that QS regulates an array of different cellular processes in bacteria. How marine bacteria interact with microalgae is no exception to this. 

Two examples of this are the bacteria *Phaeobacter gallaeciensis* and *P. inhibens*. Both are members of the Roseobacter clade and produce tropodithietic acid and roseobacticides, the latter being a unique class of algicidal chemicals. Compounds released during microalgae decomposition upregulate the biosynthesis of roseobacticides in *P. gallaeciencis* [61,62], thus turning on an ‘algicidal switch’ in the bacterium. Subsequent research confirmed that the biosynthesis of both, roseobacticides and tropodithietic acid, is also regulated by quorum sensing, specifically OH-C10-AHL [63,64]. Furthermore, along with being an antibiotic, tropodithietic acid inhibits diatom growth [40]. Therefore, in certain microalgae-associated bacteria, quorum sensing regulates the production of multiple algicidal substances. 

The concept of QS-regulated algicide production is by no means an isolated example. A prodigiosin-like pigment, isolated from a marine bacterium of the *Hahella* genus, inhibits the growth of harmful algae such as *H. akashiwo*. Researchers were able to downregulate the production of this algicidal pigment by inhibiting QS by treating *Hahella* cultures with ß-cyclodextrin (a compound that binds to AHLs and therefore prevents QS) [65,66]. Similar results were also reported by Chi and coworkers, who showed that the algicidal activity of a *Ponticoccus* strain decreased upon ß-cyclodextrin treatment [67]. The algicidal Gram-positive bacterium *Bacillus subtilis* JA, isolated from a microalga phycosphere, exhibited increased algicidal activity when supplemented with ComX, a downstream regulator of AI-2, implying that AI-2-mediated QS regulates the algicidal activity of this bacterium [68]. In a final example, Guo and colleagues used homologous recombination to show that the production of two algicides, 3-methyl indole, and the cyclic dipeptide cyclo(Gly-Phe), is regulated by AHL-mediated QS in an algicidal *Aeromonas* species [69]. Although this bacterium was isolated from freshwater, as was the corresponding cyanobacteria and green algae tested, this interaction is particularly interesting, as it appears QS positively regulates 3-methyl indole biosynthesis, while negatively regulating cyclo(Gly-Phe) biosynthesis [69]. This study also highlights a trend in the acyl chain length of AHLs in marine systems: The AHLs identified from this freshwater bacterium by Guo et al. were short, such as C4-AHL and C6-AHL, whereas AHLs in a marine system tend to be much longer, typically C10-AHL or longer [25,40]. It is possible that longer chain AHLs are better suited to marine environments, and indeed, in marine biofilms, they appear to be less susceptible to hydrolysis induced by diel fluctuations in pH (owing to photosynthesis) [70]. Daily fluctuations in AHLs are not known in planktonic scenarios, although for other phycosphere metabolites, diel oscillations do occur [71]. Overall, much in the way that QS regulates pathogenicity in human infections, there is strong evidence to suggest a similar function within the phycosphere.

## 4. QS Signals Involved in Nutrient Acquisition and Recycling and Phycosphere Structure

The microbial loop describes a stage of the marine carbon pump, which is characterized by the cyclic flow of nutrients between microbial heterotrophs and autotrophs. Much of the biochemical activity attributed to the microbial loop occurs within the phycosphere, and recent research implies that QS signals help define the microbial makeup of the phycosphere itself [72,73,74]. 

Marine phytoplankton and bacteria have evolved strategies to deal with the unique challenges of acquiring nutrients in the oceans, such as iron and phosphorous [75,76]. Research suggests that quorum sensing can regulate the process of acquiring these nutrients, as shown below.

Dimethylsulfoniopropionate (DMSP) is a ubiquitous molecule produced in large amounts by microalgae [77,78], and due to its carbon backbone and sulfur atom, is an important building block of the sulfur and carbon cycles. Supplying DMSP as a carbon source for the bacterium *Ruegeria pomeroyi* upregulates oxo-C14-AHL biosynthesis (compared to supplying the bacterium with propionate), suggesting that AHL-based QS and DMSP metabolism are coupled [79]. Landa et al. provided transcriptomic evidence from an elegant tripartite system to support this result. In this system, *R. pomeroyi* transcripts were quantified during co-culture with a DMSP-producing dinoflagellate and a DMSP-non-producing diatom. As the experiment progressed, bacteria cell densities remained steady, while the microalgae population shifted from predominantly dinoflagellate species to diatom species. This shift correlated with a change in bacterial metabolism, including the downregulation of AHL biosynthesis and perception, as the bacterium adapted to new primary producers [80]. Thus, there is both metabolomic and transcriptomic evidence that AHL biosynthesis is dictated by the nutrients presented to *R. pomeroyi*.

One of the key outcomes of the microbial loop is the production of extracellular particulate organic matter, which ultimately becomes marine snow as these particles begin to sink. Bacteria are known to capitalize on these nutrient-rich particles, and research has investigated if QS plays a role during the metabolism of these nutrients [22]. Recently, Su and coworkers isolated bacteria from marine particles, and found a *Ruegeria mobilis* strain, which responded to exogenous AHL application by altering the excretion of extracellular enzymes, including lipid- and carbohydrate-degrading enzymes [81]. This research built upon earlier work, which also showed hydrolytic enzyme production, including lipases, is regulated by AHLs in marine particle bacteria [82,83]. These studies have therefore shown a strong correlation between AHL signaling and extracellular enzyme activity, suggesting that QS plays a role in breaking down marine particles and nutrient recycling. In both works conducted by Jatt, Hmelo and respective coworkers, phosphatase activity was also regulated by QS, echoing results reported by Van Mooy et al. [84], who tracked alkaline phosphatase activity in the phycosphere of the marine cyanobacterium Trichodesmium. They showed that alkaline phosphatase production in these phycospheres is positively regulated by AHL QS, but negatively regulated by AI-2 QS. This research suggests that certain marine bacteria utilize AHLs and AI-2 to regulate digestive enzymes’ production in the microenvironments surrounding microalgae or marine snow. 

Iron is often particularly scarce among marine ecosystems, and marine bacteria utilize iron chelators to improve iron uptake [85]. Interestingly, both PQS and tetramic acids are known to chelate iron [17,86], so it is tempting to hypothesize that these compounds might also aid in iron acquisition. While a similar process has been observed for PQS in other environments [87], the role of tetramic acids and PQS as chelators in marine systems is yet to be elucidated. Indeed, despite the research summarized above, we are only beginning to understand the role of QS in nutrient acquisition and the community architecture of marine microbial communities.

## 5. Cyanobacteria: A Special Case of QS-Producers

Cyanobacteria are unique microalgae, being both phytoplankton and QS signal producers. AHLs from samples of microbial mats in alpine lakes were identified by Bachofen and Schenk [88], who hypothesized that the source of these AHLs was cyanobacteria, the predominant organisms of the samples. Later studies supported this viewpoint, in which AHLs were isolated from axenic cyanobacterial cultures of *Anabaena* [89], *Gloeothece* [90] and *Microcystis aeruginosa* [91], and provided evidence that AHLs contribute to the formation of microcolonies [91]. The physiological effects of AHLs on the diazotrophic species *Anabaena* was also explored by Romero and coworkers [92], who demonstrated that oxo-10-AHL had striking algicidal properties. Furthermore, the authors demonstrated that a range of AHLs inhibited nitrogen fixation, however, no clear mechanism was demonstrated. Previous work by Romero and coworkers also identified an AHL-degrading acylase in the proteome of *Anabaena* [89], suggesting that this cyanobacterium has evolved a means to modulate AHL concentrations. 

Little work has been conducted regarding the effects of AQs on cyanobacteria, although Long and coworkers showed that growth and motility in marine strains of *Synechocuccus* is inhibited by 2-pentyl-quinolone [48]. In comparison to investigations into the effects of QS signals on other microalgae, our current knowledge of their effects on cyanobacteria is limited. Indeed, much of the work on cyanobacteria has so far focused on freshwater species, with less known about their marine counterparts. 

## 6. Can Microalgae Modulate Quorum-Sensing?

Given many QS molecules often have a deleterious effect on microalgae, it would be natural to assume that microalgae have evolved the ability to defend themselves in some way. This has been observed for other algicidal compounds [93]. Therefore, can microalgae ‘intercept’ quorum-sensing signals? 

How microbes interfere with quorum signals is an area of intense research [94], and is well documented among marine bacteria [33,95,96]. Different mechanisms are employed by these bacteria to do this, which can be divided into (1) interfering with QS circuitry by producing inhibitory molecules, thus inhibiting the ability of a bacterium to sense or synthesize QS signals, and (2) producing enzymes which degrade QS signals (often called quorum quenching). If such bacteria were to live within an algal phycosphere, one could hypothesize that the host alga would benefit from this process. For instance, an alga could ensure that growth-inhibiting tetramic acids or AQs do not reach damaging concentrations within the phycosphere by fostering mutualistic relationships with bacteria that can modify AQs [97] or produce quorum-quenching acylases. Such a mutualism remains speculative in this case, but a similar scenario, in which bacteria produce B vitamins in exchange for carbohydrates produced by the microalga, is well documented [7,98,99,100,101]. In any case, QS interference is well documented in marine bacteria, but the wider role played by QS interference in planktonic environments is still unclear.

Compared to marine bacteria, very little is known about the QS interference abilities of microalgae, and what is known predominantly relates to green algae and cyanobacteria. For example, a range of compounds interfering with QS have been isolated from marine cyanobacteria [102]. As for green algae, a seminal 2004 study showed that species of *Chlorella* and *Chlamydomonas reinhardtii* excreted a range of compounds interacting with quorum sensing-dependent fluorescence in reporter strains, specifically those responding to long-chain AHLs and AI-2 QS signals [103]. These experiments found that compounds excreted by *C. reinhardtii* in fact induced quorum sensing. The authors were able to extract and purify active constituents from organic extracts from *C. reinhardtii*, suggesting that QS induction was due to excreted metabolites. In a following study, Rajamani and coworkers ascertained that these QS-inducing molecules possess a lactone moiety, but could not identify any AHLs excreted by the alga [104]. These two studies also simultaneously identified a quorum-quenching effect, which the authors attributed to the activity of lactonase enzymes in *C. reinhardtii* [103,104]. Although these studies focused on freshwater species, later work with both marine and freshwater microalgae (including green microalgae) reported several strains of microalgae interfering with AHL- and AI-2-based QS in multiple reporter systems [105]. Future works are necessary to identify these quorum-quenching lactonases and QS mimic compounds. 

The structure of one QS mimic from *C. reinhardtii* was derived [106]. Lumichrome, a derivative of riboflavin, was found to induce QS via stimulation of the LasR gene, which responds to long-chain AHLs such as oxo-12-HSL. It is important to note that based on the chemical structure of lumichrome, it should not be sensitive to degradation by lactonases, meaning the other mimics detected in *C. reinhardtii* extracts are different to lumichrome.

The diatom *Nitzschia cf pellucida* does degrade AHLs. Diatoms are well known to produce a diverse array of halogenated metabolites. Syrpas and colleagues observed the stepwise halogenation of oxo-AHLs by the diatom, leading to hydrolysis of both the acyl and lactone moieties [107]. This result implies that diatoms can degrade AHLs, thus modulating any algicidal effects and interfering with bacterial QS-derived physiology. While *N. cf pellucida* is a benthic diatom, it is plausible that planktonic diatoms conduct similar biochemistry. Intriguingly, a similar mode of QS detoxification via halogenation has also been observed in the marine bacterium *Microbulbifer* sp. HZ11 [108]. *Microbulbifer* sp. HZ11 does not conduct alkyl quinolone-mediated QS, yet it was able to detoxify AQs via bromination. The resulting bromo-quinolones were less toxic to *M.* sp. HZ11 than their precursors, while displaying increased toxicity to other bacteria. Ritzmann and coauthors also identified the haloperoxidase gene responsible, and found similar genes in other marine prokaryotes and eukaryotes, suggesting that this behavior occurs in other marine organisms. 

As shown above, QS interference in microalgae remains an area of untapped potential, particularly given the well-known history of QS interference from macroalgae [109,110]. Little is known about how these microbes interfere with AQ- or AHL-mediated QS. Furthermore, while diatoms may have the ability to degrade AHLs [107], there are no known AHL receptors in microalgae whatsoever.

## 7. Conclusions

Since Azam and co-authors coined the term ‘microbial loop’ [111], we have gained a more in-depth view into bacteria–phytoplankton interactions, thanks to ongoing genetic, physiological and biochemical advances. These advances illustrate that the microbial loop should not be considered as a loop per se, but rather as a ‘microbial network’, in which a multitude of different connections exist between bacteria and phytoplankton. Quorum- sensing signals are one of these connections. They modulate phytoplankton behavior and proliferation in ways that we have begun to understand in recent years (summarized in Figure 2). Furthermore, many of the AQs discussed above are not known to conduct quorum sensing in their respective bacteria at all. Thus, many compounds typically classified as QS signals are in fact better regarded as algicides (or as antimicrobials [112]) with structural similarity to known QS signals. 

Through interdisciplinary research, our understanding of QS in the phycosphere has revealed the intracellular mechanisms by which QS inhibits microalgae growth and structures the phycosphere as a whole [57,58,73]. The surprising diversity of QS signals from some marine bacteria also offers the tantalizing possibility of even more complex interactions [25]. However, this review also highlights the importance of examining QS signals beyond AHLs, as shown by the striking effects of AQs on microalgae. Furthermore, as the production of algae for agri- and aqua-cultural processes continues to grow worldwide, the presence of QS signals in algal photobioreactors should be considered as an important factor in efforts to improve the output of these vessels (as emphasized by recent research [29,113]). Finally, there is nascent evidence to suggest that microalgae have developed methods to modulate QS signals, but more research is required to ascertain the extent and importance of these observations. Thus, our understanding of the ways in which quorum sensing dictates marine microbial life demonstrates how a single molecule can direct life in the phycosphere and its global ramifications. 

## Figures and Tables

**Figure 1 microorganisms-09-01391-f001:**
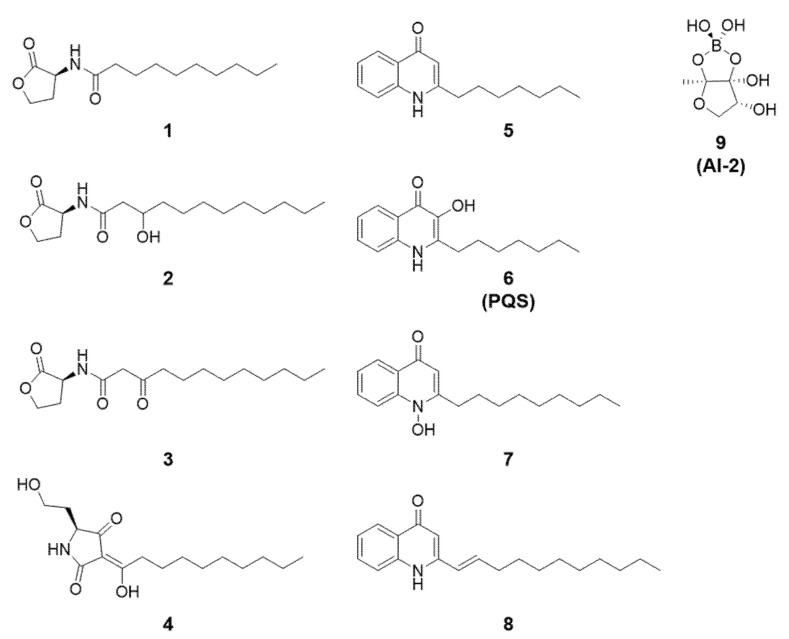
Representative compounds of relevant quorum-sensing signals. 1: C10-AHL, 2: OH-C12-AHL, 3: oxo-C12-AHL, 4: tetramic acid TA12, formed by spontaneous cyclization of oxo-C12-AHL, 5: heptyl-quinolone, 6: pseudomonas quorum signal (PQS) 2-heptyl-3-hydroxy-quinolone, 7: nonyl-quinolone-N-oxide, 8: 1′E-undecenyl-quinolone, 9: furanosyl borate diester, AI-2.

**Figure 2 microorganisms-09-01391-f002:**
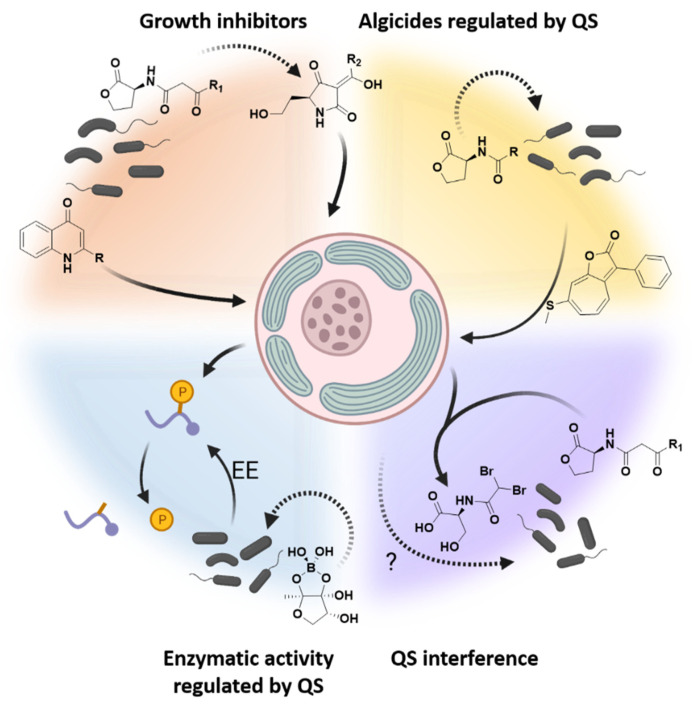
Graphical summary of the algae–bacteria interactions dictated by quorum-sensing signals in marine planktonic systems. These QS signals can be directly algicidal, as shown in the orange sector, such as alkyl-quinolones [57] and tetramic acids [43], which inhibit growth via photosynthesis inhibition. Quinolones and tetramic acids also arrest cell-cycle activities and several other metabolic processes in microalgae [42,58]. AHL-mediated quorum sensing regulates the production of other algicidal compounds (yellow sector), such as the roseobacticide shown [64]. Quorum sensing regulates extracellular enzymatic activity in marine bacteria (blue sector), allowing these microbes to capitalize on encounters with nutrient hotspots, such as by liberating phosphates from organic molecules (blue curves with orange P) on marine particles and in phycospheres [80,84]. Research is also hinting at the ways by which microalgae interfere with quorum sensing, shown in the lilac sector [104,107]. Dotted lines represent indirect actions of QS signals, while full lines represent direct effects. EE: extracellular enzymes. Created with BioRender.com.

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
