# Peer review of "How Do Quorum-Sensing Signals Mediate Algae–Bacteria Interactions?"

_microorganisms, 2021, doi:10.3390/microorganisms9071391_

Round 1
Reviewer 1 Report
I really enjoyed reading this excellent review paper on QS in context with algae-bacteria interactions. The topic is comprehensively covered and timely.
A couple of questions and curious thoughts popped up that may deserve attention through minor revision. Adressing these points is not a must but may strengthen this paper as other readers might think along similar lines.
- how do the largely negative effects of QS signals on microalgae happen on a mechanistic level? Do algae have extracellular QS receptors and if so, are the observed efects rendered by downstream intracellular responses? Do we know anything about the causative molecules in this case? Or is the microalgal cell wall permeable to QS signals, and if so, is it thus the QS molecule itself affecting algal physiology?
- You mention LC/IC-50 values of certain AQs (line 139, 147). These values are in the micromolar range. For a phycosphere environment, these are very high concentrations I would argue. Are these concentrations ever reached by phycosphere-associated bacteria? Can you elaborate on this if known at all?
- In your discussion whether microalgae can modulate QS (Chapt. 6, l. 254) I was wondering if the phycosphere pH during day and night as a result of photosynthesis may change to such extent that lactone hydrolysis may occur? As QS signals are quite susceptible to oxidation and pH, both oxygen and carbon dioxide production may have drastic effects on QS signals in phycospheres.
Otherwise, I'm happy with this review as is, and recommend publication without further needs of revision.
Author Response
Dear reviewer,
Thank you for your constructive and meaningful feedback. My responses to your reviews are discussed below:
R1: How do the largely negative effects of QS signals on microalgae happen on a mechanistic level? Do algae have extracellular QS receptors and if so, are the observed effects rendered by downstream intracellular responses? Do we know anything about the causative molecules in this case? Or is the microalgal cell wall permeable to QS signals, and if so, is it thus the QS molecule itself affecting algal physiology?
A: Mechanistically, QS signals which inhibit photosynthesis and respiration permeate into the cell, and directly interact with elements of chloroplasts and mitochondria. Other responses, particularly those of the transcriptome, do appear to be due to some level of regulation, but what exactly we don’t know. I have made this clearer in the manuscript (L164).
There are no known QS receptors in algae, nor plants (to find one would indeed be a great discovery). I have added this information at line 306.
R1: You mention LC/IC-50 values of certain AQs (line 139, 147). These values are in the micromolar range. For a phycosphere environment, these are very high concentrations I would argue. Are these concentrations ever reached by phycosphere-associated bacteria? Can you elaborate on this if known at all?
Indeed these are high concentrations. Unfortunately, we don’t have any data regarding phycosphere concentrations of these metabolites. Modelling data suggest that metabolite concentrations on the exterior of microalgae can be several orders of magnitude higher than their bulk water concentrations (for further information see Seymour et al 2010 doi: 10.1126/science.1188418, Seymour et al 2017 doi:10.1038/nmicrobiol.2017.65). I have opted to leave this discussion out of the manuscript.
R1: In your discussion whether microalgae can modulate QS (Chapt. 6, l. 254) I was wondering if the phycosphere pH during day and night as a result of photosynthesis may change to such extent that lactone hydrolysis may occur? As QS signals are quite susceptible to oxidation and pH, both oxygen and carbon dioxide production may have drastic effects on QS signals in phycospheres.
This is an excellent observation. I have added information to the manuscript at line 200 covering this topic.
Thank you for taking the time to review my manuscript!
Reviewer 2 Report
The review is very interesting and it will be useful for many researchers. Indeed, algae-bacteria interactions are very complex, and the review brings many details on this topic. However, some references may be added in order the topic covers algae-bacteria interactions in different structures.
Please add :
- Cirri & Pohnert (2019) New Phytologist 223: 100–106
- Wang et al (2018) Toxins 10, 257; doi:10.3390/toxins10070257
- Zhang et al (2018) Appl Microbiol Biotechnol 102:5343–5353: quorum sensing in granular sludge
- Malik et al (2020) Microbiology 89, No. 6, pp. 778–788 : quorum sensing on macroalgae
- Carvalho et al (2017) J Appl Phycol 29:789–797 : quorum sensing on macroalgae
Line 62 : add at least one reference for biofilm (74 ?; see also introduction in Das et al (2019) Bioresource Technology 294, 122138)
Author Response
Dear reviewer,
Thank you for your constructive and meaningful feedback. My responses to your reviews are discussed below:
R2: Line 62 : add at least one reference for biofilm (74 ?; see also introduction in Das et al (2019) Bioresource Technology 294, 122138)
I have added a references here which discusses the effects quorum sensing has on bacterial physiology, including biofilm formation (Line 61)
R2: Please add
- Cirri & Pohnert (2019) New Phytologist 223: 100–106
Added at Line 90 - Wang et al (2018) Toxins 10, 257; doi:10.3390/toxins10070257
I have opted not to cite this publication. Despite the importance of this meaningful publication for harmful algal blooms, Wang et al made only tangential relations between quorum sensing and the effect of these bacteria on Gambierdiscus. Further work is necessary to provide evidence that quorum sensing is actually involved in the interaction between bacterium and Gambierdiscus. - Zhang et al (2018) Appl Microbiol Biotechnol 102:5343–5353: quorum sensing in granular sludge
This reference added at line 333 - Malik et al (2020) Microbiology 89, No. 6, pp. 778–788 : quorum sensing on macroalgae
This reference added at lines 83 and 333. I was not aware of this publication, thank you for bringing it to my attention. - Carvalho et al (2017) J Appl Phycol 29:789–797 : quorum sensing on macroalgae
This reference added at line 302.
Thank you for taking the time to review my manuscript!